# An Expression Tree Decoding Strategy for Mathematical Equation Generation

**Wenqi Zhang**[1], **Yongliang Shen**[1†] , **Qingpeng Nong**[2], **Zeqi Tan**[1]
**Yanna Ma**[3], **Weiming Lu**[1†]
[1]College of Computer Science and Technology, Zhejiang University
[2]Zhongxing Telecommunication Equipment Corporationy
[3]University of Shanghai for Science and Technology
{zhangwenqi, luwm}@zju.edu.cn

## Abstract

Generating mathematical equations from natural language requires an accurate understanding of the relations among math expressions. Existing approaches can be broadly categorized into token-level and expression-level generation. The former treats equations as a mathematical language, sequentially generating math tokens. Expression-level methods generate each expression one by one. However, each expression represents a solving step, and there naturally exist parallel or dependent relations between these steps, which are ignored by current sequential methods. Therefore, we integrate tree structure into the expression-level generation and advocate an expression tree decoding strategy. To generate a tree with expression as its node, we employ a layer-wise parallel decoding strategy: we decode multiple independent expressions (leaf nodes) in parallel at each layer and repeat parallel decoding layer by layer to sequentially generate these parent node expressions that depend on others. Besides, a bipartite matching algorithm is adopted to align multiple predictions with annotations for each layer. Experiments show our method outperforms other baselines, especially for these equations with complex structures.

## 1   Introduction

Generating corresponding mathematical equations and solutions from text is important for a range of tasks, such as dialogues, question answering, math word problem (MWP), etc. It necessitates an accurate comprehension of semantics and the complex relations between mathematical symbols.

We investigate the existing approaches from two perspectives: the generation is at the **token-level or expression-level**, and the order is based on **sequence or tree structure**. Firstly, sequence-to-sequence methods (seq2seq in Figure 1) (Wang et al., 2017, 2018; Chiang and Chen, 2019; Li et al.,

---

[†]Corresponding author.

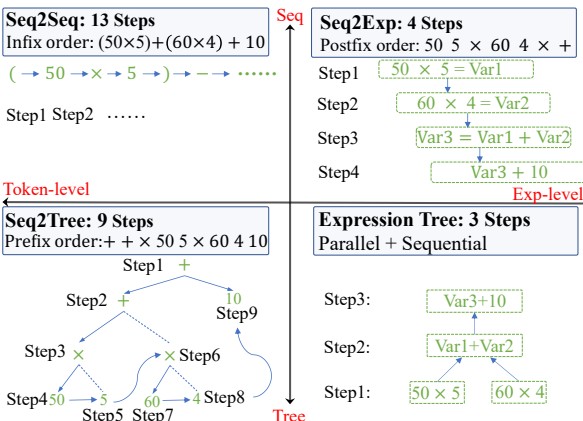

Figure 1: Four equation generation methods: Seq2Seq generates mathematical tokens one by one at the token-level; Seq2Tree generates tokens using prefix order. In contrast, Seq2Exp generates expressions one by one at expression-level. Our expression tree decoding strategy predicts multiple math expressions in parallel and forms a tree, with the minimum decoding steps.

2019) have considered mathematical symbols as a special kind of language (i.e., mathematical language) and employ sequential generation for the equation. These methods belong to the token-level sequential generation. Then a great deal of work (Xie and Sun, 2019; Zhang et al., 2020; Patel et al., 2021a; Li et al., 2020; Zhang et al., 2022c; Shen et al., 2022b) has proposed a tree-order decoding process (seq2tree in Figure 1) at the token-level. This process considers it as an equation tree generation and predicts pre-order tokens one by one.

Recently, some researchers have explored expression-level approaches for mathematical equation generation, including (Kim et al., 2020; Cao et al., 2021; Jie et al., 2022; Zhang and Moshfeghi, 2022; Zhang et al., 2022a). These approaches mostly place emphasis on generating a mathematical expression step by step (seq2exp), rather than a token. These seq2exp methods belong to a sequen-

tial generation at expression-level.

However, it is imperative to recognize that each mathematical expression represents a problem-solving step, and there inherently exists a parallel or dependent relation among these steps. The existing seq2exp approach may struggle to capture these relations since they only produce expressions in sequence. Therefore, there is a pressing need for a versatile decoding strategy capable of simultaneously generating independent expressions in parallel at one step, while sequentially producing expressions that depend on others step by step.

Based on this belief, we propose an expression tree decoding strategy by combining the seq2exp with a tree structure at the expression level. Differing from the prior seq2tree, each node in this tree represents an expression, rather than a token. To construct an expression-level tree, we generate multiple expressions in parallel at each step. These expressions are independent to each other and act as the leaf node in the expression tree. Those expressions depend on others, they act as parent nodes and are sequentially generated based on their child nodes. As shown in Figure 1, the two expressions $(50 \times 5, 60 \times 4)$ are completely independent and generated in parallel at Step 1. The third expression depends on the first two, forming an expression tree. It not only empowers the model to exploit the inherent structure of the equation but also shortens its decoding path (the minimum steps in Figure 1).

To achieve this, we design a layer-wise parallel decoding strategy. At each decoder's layer, it can generate multiple independent expressions in parallel and then proceeds to the next layer, and repeat parallel prediction layer by layer. This layer-wise parallel decoding process ensures that these independent expressions are produced in parallel, while these expressions depending on others are sequentially generated layer by layer, eventually constructing the entire expression tree.

Besides, to decode multiple expressions in parallel, we take inspiration from query-based object detection (Carion et al., 2020; Jaegle et al., 2021; Li et al., 2023). Similar to detecting multiple visual objects by queries, we also utilize queries to identify multiple mathematical relations for expression generation in parallel. Lastly, we adopt a bipartite matching algorithm for loss calculation between multiple predicted expressions and the label.

Cao et al. (2021) shares some similarities with us but the challenges and approaches are different.

Cao et al. (2021) perform a bottom-up structure to extract multiple equations (e.g., x+y=3, y-2=4), whereas our method considers how to predict multiple expressions in parallel for a complex equation at each step (e.g., output $50 \times 5$, $60 \times 4$ simultaneously for $50 \times 5 + 60 \times 4$). Besides, Cao et al. (2021) enumerates all possible expression combinations but we first introduce bipartite matching to achieve parallel prediction.

Our contributions are threefold:

- We introduce an expression tree decoding strategy by combining seq2exp with tree structure. It considers the dependent or parallel relation among different expressions (solving steps). To the best of our knowledge, this is the first effort to integrate query-based object detection techniques with equation generation in the literature.
- We design a layer-wise parallel decoding process to construct an expression tree. It predicts multiple independent expressions in parallel and repeats parallel decoding layer by layer. Besides, we employ bipartite matching to align predicted expressions with labels.
- To assess our method, we evaluate on MWP task and outperform prior baselines with higher accuracy and shorter steps.

By aligning the decoding process with the inherent structure of the equation, our approach paves the way for more intuitive, efficient equation generation. Moreover, it provides insights that could be applicable to many other structured tasks.

## 2 Related work

In advancing toward general-purpose AI, dependable reasoning remains imperative. The quest for human-equivalent reasoning has been rigorously explored in domains including NLP (Kojima et al., 2022), RL (Zhang et al., 2022b), and Robotics (Zhang et al., 2021). Recently, leveraging the planning and reasoning capabilities of LLMs paves the way for the development of numerous intelligent applications (Wei et al., 2022; Shen et al., 2023; Zhang et al., 2023b). Accurate generation of mathematical equations is an important manifestation of reasoning abilities, which has been extensively investigated in a plethora of NLP tasks, e.g., Math Word Problems (Wang et al., 2017; Ling et al., 2017; Xie and Sun, 2019; Wang et al., 2022a), Question Answering (Yu et al., 2018; Wu et al.,

2020b), Dialogue (Bocklisch et al., 2017; Wu et al., 2022, 2023), etc. These tasks necessitate an accurate understanding of the semantics within the text as well as mathematical symbols.

**Token-level Generation** Mathematical equation was treated as a translation task from human language into the mathematical token (symbol) (Wang et al., 2017; Chiang and Chen, 2019). Many seq2seq-based methods were proposed with an encoder-decoder framework. Li et al. (2019) introduced a group attention to enhance seq2seq performance. Lan et al. (2021) utilized a transformer model for equation generation. Except for seq2seq methods, some researchers (Liu et al., 2019a; Xie and Sun, 2019; Zhang et al., 2020, 2022c) studied the decoding structures and proposed a tree-based decoder using prefix sequence. Wu et al. (2020a, 2021); Qin et al. (2021); Yu et al. (2021) introduced mathematical knowledge to solve the complex math reasoning. Liang et al. (2021b) improved accuracy by knowledge distillation between the teacher and student. Li et al. (2021) proposed a prototype learning through contrasting different equations. Shen et al. (2022a); Liang et al. (2022) also used contrastive learning at the semantic and symbolic expression levels. Yang et al. (2022b) improved the interpretability by explicitly retrieving logical rules. These generative approaches were token-level generation in infix order or prefix order.

**Expression-level Generation** Expression-level generation has opened up a new perspective for math solving. Kim et al. (2020) proposed an expression pointer generation. Cao et al. (2021) introduced a DAG structure to extract two quantities from bottom to top. Jie et al. (2022); Wang and Lu (2023) further treated this task as an iterative relation extraction to construct an expression at each step. Zhang and Moshfeghi (2022) treated expression generation as a program generation and execution. Besides, Lee and Kim (2023); Zhang et al. (2023a); He-Yueya et al. (2023); Zhu et al. (2022) harness the capabilities of LLMs and prompt engineering to bolster mathematical reasoning under the few-shot setting. These methods treat equation generation as a multi-step expression generation and achieve impressive performance. However, these methods generate only one expression per step using pre-defined order, which may potentially impede the model's acuity in accurately understanding mathematical logic. In contrast, our method generates multiple expressions in parallel per step.

## 3 Methodology

### 3.1 Overview

The task is to generate a complete equation based on the problem description. The generation process contains two vocabularies: number and operator vocabulary ( $V_{op}=\{+,-,\times,\div,\cdots\}$ ). The number is either from the original text or the math expression results from previous steps.

Similar to object detection, where queries are utilized to detect multiple objects, we also feed multiple learnable queries to identify possible math relations. As shown in Figure 2, a standard decoder and multiple queries are adopted to generate candidate expressions at each layer. To construct an expression tree, we must consider two structures simultaneously: parallel and sequential. For these expressions that have no dependent relations, we employ a parallel strategy to generate them. Conversely, for these expressions that depend on others, we generate them layer by layer (§ 3.2). We also provide detailed cases in Figure A2 for decoding. When training, we utilize a bipartite matching algorithm to align the multiple predicted expressions with the label set for loss calculation (§ 3.3).

### 3.2 Layer-wise Parallel Decoding

We devise a problem encoder, a decoder, and multiple learnable queries, where the query is used to identify a specific mathematical relation and then produce a candidate expression at each layer.

**Problem Encoder** Given a text description $X$ with $N_n$ number words, we adopt the pre-trained language model (Devlin et al., 2019; Liu et al., 2019b) to obtain the contextualized problem representations $P$. We obtain number vocabulary $V_n = \{e_n^i\}_{i=1}^{N_n}$ from $P$, which denotes the embeddings of the $N_n$ number tokens from text. In addition, we randomly initialize the learnable embedding for each operator and a *None* label $V_{op}=\{e_{op}^j\}_{j=1}^{N_{op}+1}$.

**Learnable Query** The decoder is designed for extracting all candidate expressions in parallel based on problem representation. Firstly, we design learnable embeddings as query $Q = \{q_i\}_{i=1}^{K}$, where $K$ means the number of the query. As shown in Figure 2, the $K$ queries are firstly fed into the decoder and then are utilized to predict $K$ possible expressions at each layer.

Specifically, the decoder is the standard transformer decoder which contains a stack of identical layers. At the $l$-th layer, the problem embeddings $P^{l-1}$ and the query embeddings $Q^{l-1}$ from the

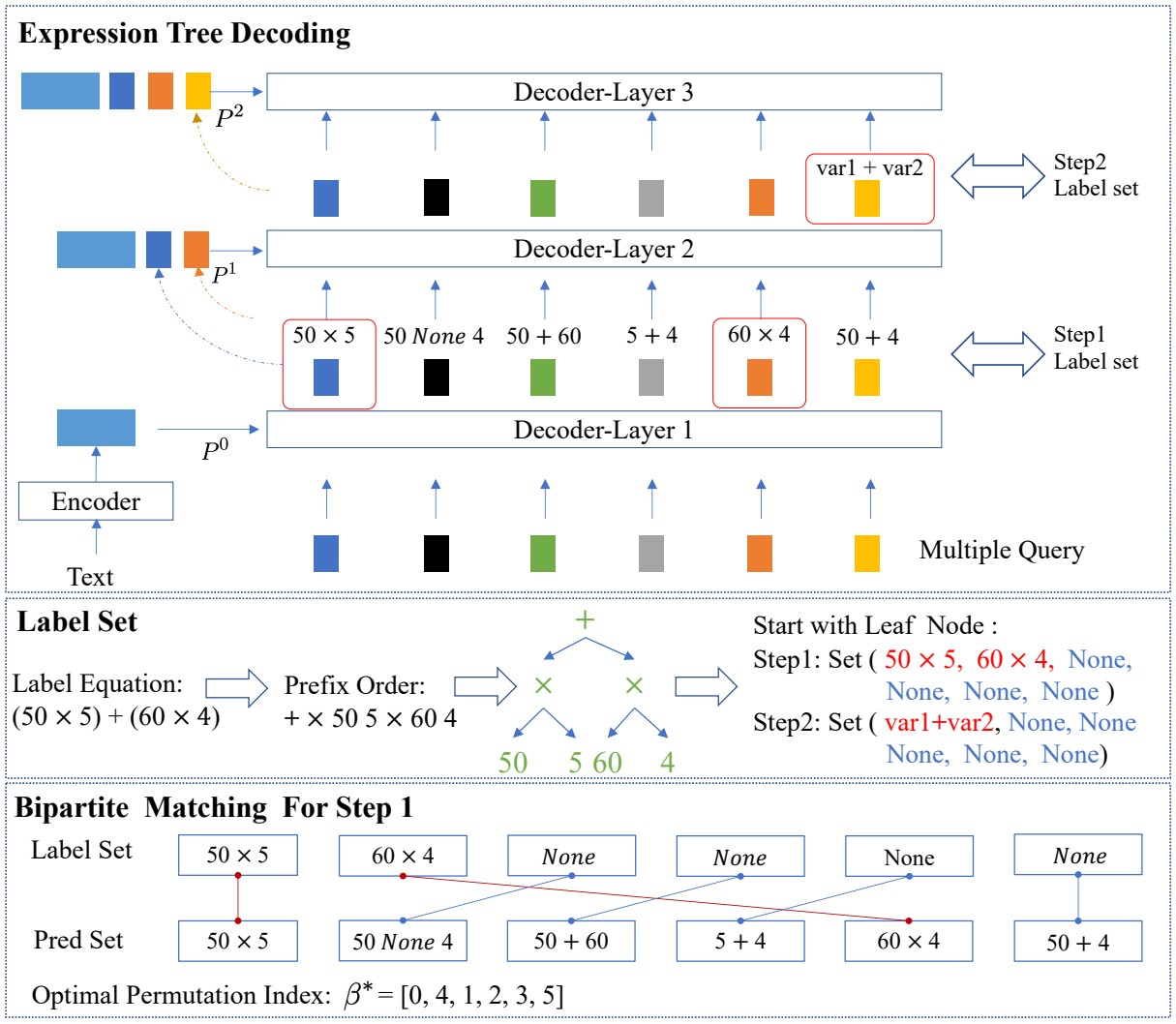

Figure 2: We propose an expression tree decoding strategy by layer-wise parallel decoding. During training, we feed six queries into the decoder, where each decoder's layer generates six mathematical expressions. Then, we transform the original equation label into multiple label sets and employ a bipartite matching algorithm to align the label sets with the six predicted expressions for loss calculation. Thereafter, we update the problem representation using valid expressions and feed it into the next decoding layer. The whole process forms an expression tree.

previous layer are fed into the current decoder's layer as inputs, and then interact with each other by self-attention and cross-attention mechanism:

$$Q^l = \text{Decoder-Layer}^l(Q^{l-1}; P^{l-1}) \qquad (1)$$

where $Q^l$ means $K$ query embeddings at $l$-th layer.

**Parallel Decoding** After obtaining $K$ query vectors, we use them to predict $K$ expressions in parallel, each of which contains one operator and two operands $(\underline{l}, \underline{op}, \underline{r})$. Firstly, we use query to calculate the operator and operands embedding:

$$s_i^l, s_i^r, s_i^{op} = \text{MLP}^{l,r,op}(q_i), q_i \in Q^l \qquad (2)$$

where $q_i$ denotes the $i$-th query vectors in $Q^l$. Then,

we predict three distributions as follows:

$$P_i^{l,r}(*) = \text{Softmax}(s_i^{l,r}e_n), \forall e_n \in V_n \qquad (3)$$
$$P_i^{op}(*) = \text{Softmax}(s_i^{op}e_{op}), \forall e_{op} \in V_{op} \qquad (4)$$

where $e_n$ and $e_{op}$ represent the number and operator embedding in vocabulary respectively. $P_i^l(*)$, $P_i^r(*)$, and $P_i^{op}(*)$ denotes the three distributions for two operands and one operator. Lastly, we calculate the embedding for this $i$-th expression:

$$var_i = \text{MLP}^n([s_i^{op}; s_i^l; s_i^r; s_i^l \circ s_i^r]) \qquad (5)$$

We totally predict $K$ independent expressions from $K$ queries at this layer. Then we continue to the next layer for these expressions depending on previous results (e.g., $var1 + var2$ in Figure 2).

**Layer by Layer Prediction** The $K$ predicted expressions contain $K_1$ valid expressions and $K_2$ invalid expressions. An invalid expression implies that the operator in $(l, op, r)$ is predicted to *None*. We will discuss it in section (§ 3.3). First, we concat all valid expression embeddings with the problem representations $P^{l-1}$ for the next layer:

$$P^l = \text{MLP}^u([P^{l-1} \oplus var_1 \oplus var_2 .... \oplus var_{K_1}]) \quad (6)$$

where $var_1$, $var_2$, ...., $var_{K_1}$ means $K_1$ new generated expression embeddings in $l$-th layer. $\oplus$ means the concatenation of the vectors. Besides, we also update the number embedding: $V_n^l = V_n^{l-1} \cup var_1 \cup var_2 .... \cup var_{K_1}$ using $K_1$ new expression embeddings as new number embeddings.

As shown in Figure 2, we proceed to the next layer of decoding using $P^l$ and $Q^l$, as Equation 1:

$$Q^{l+1} = \text{Decoder-Layer}^{l+1}(Q^l; P^l) \quad (7)$$

At layer $l+1$, we still decode $K$ expressions in parallel and continue to the next layer. If all predicted expressions are invalid, i.e., $K$ operators are classified as $None$, it signifies the equation generation is complete.

### 3.3 Loss For Parallel Decoding

As mentioned before, each decoder's layer generates multiple mathematical expressions. However, the annotations of equations in the dataset are usually serial (e.g., "50", "×", "5", "+", "60", "×", "4"), and it is crucial to design a method to utilize these annotations for training parallel decoding.

To this end, we first convert the original equation annotations into multiple label sets, each set comprising $K$ mathematical expressions. Then, bipartite matching is employed to align the $K$ predicted expressions with the $K$ mathematical expressions in the label set to compute model loss.

**Label Set** As shown in Figure 2, we initially convert the label equations from infix order to prefix order, thus forming an equation tree. Starting from the leaf nodes, we iterative gather two leaves and their parent node into a label set for each step, eventually producing multiple label sets (e.g. set1 = $\{50 \times 5, 60 \times 4\}$, set2 = $\{var1 \times var2\}$). Each label set contains several non-dependent expressions that can be generated in parallel. Each label set is also padded with a specific label $None$ to ensure all sets contain $K$ elements. We provide two detailed cases for this process in Figure A2.

**Bipartite Match** For each layer, $K$ candidate mathematical expressions are predicted. We compute the function loss for the $K$ predicted expressions based on the corresponding label set, which also contains $K$ golden expressions. However, as the $K$ expressions are unordered, it is difficult to calculate the loss directly. For instance, if the label set is $\{50 \times 5, 60 \times 4\}$, and the prediction is $\{60 \times 4, 50 \times 5\}$, the loss in this case should be 0. To address this, we adopt a bipartite matching algorithm to align the two sets, i.e., align the $K$ predictions with the $K$ golden labels. As shown in Figure 2, six golden expressions align with six predicted expressions. Specifically, we denote the golden expression in the label set as $\{y_1, y_2, ..., y_K\}$, and the set of predicted expressions by $\hat{y} = \{\hat{y}_i\}_{i=1}^K$. To find an optimal matching, we search for a permutation ($\beta \in \mathcal{O}_K$) of $K$ elements with the lowest cost. As shown in Figure 2, the optimal permutation for predicted set is $[\hat{y}_0, \hat{y}_4, \hat{y}_1, \hat{y}_2, \hat{y}_3, \hat{y}_5]$. It can be formalized as:

$$\beta^* = \arg\min_{\beta \in \mathcal{O}_K} \sum_i^K \mathcal{L}_{\text{match}}(y_i, \hat{y}_{\beta(i)}) \quad (8)$$

where $\mathcal{L}_{\text{match}}(y_i, \hat{y}_{\beta(i)})$ is a pair matching cost between the golden expression $y_i$ and the predicted expression $\hat{y}$ with index $\beta(i)$. We use the Hungarian algorithm (Kuhn, 1955) to compute this pair-matching cost. Each golden expression contains two operands and one operator, i.e. $y_i = (l_i, op_i, r_i)$ and each predicted expression has three distributions, i.e. $\hat{y}_i = (P_i^l(*), P_i^{op}(*), P_i^r(*))$. We calculate $\mathcal{L}_{\text{match}}$ as follow:

$$\mathcal{L}_{\text{match}}(y_i, \hat{y}_{\beta(i)}) = -\mathbb{1}_{\{op_i \neq None\}} \left[ p_{\beta(i)}^{op}(op_i) \right. $$
$$\left. + p_{\beta(i)}^l(l_i) + p_{\beta(i)}^r(r_i) \right] \quad (9)$$

After we get the optimal matching $\beta^*$, we calculate the final loss $\mathcal{L}(y, \hat{y})$ as:

$$\mathcal{L}(y, \hat{y}) = \sum_{i=1}^N \left\{ -\log p_{\beta^*(i)}^{op}(op_i) \right. $$
$$+ \mathbb{1}_{\{op_i \neq None\}} \left[ -\log p_{\beta^*(i)}^l(l_i) \right. $$
$$\left. \left. -\log p_{\beta^*(i)}^r(r_i) \right] \right\} \quad (10)$$

We calculate the predicted loss for each decoder layer after aligning two sets. A detailed match process is provided in Figure A3.

| | Model | Test Acc. |
|---|---|---|
| **Seq2Seq / Tree** | GroupAttn(Li et al., 2019) | 70.4 |
| | GTS (Xie and Sun, 2019) | 71.3 |
| | G2T(Zhang et al., 2020) | 72.0 |
| | BERT-T(Liang et al., 2021a) | 73.8 |
| | mBERT(Tan et al., 2021) | 77.1 |
| | T-Dis†(Liang et al., 2021b) | 73.1 |
| | Prototype (Li et al., 2021) | 76.3 |
| | Textual-CL†(Shen et al., 2022a) | 78 |
| | Ana-CL (Liang et al., 2022) | 79.6 |
| **Seq2Exp** | E-pointer†(Kim et al., 2020) | 73.5 |
| | M-Tree†(Wang et al., 2022a) | 76.5 |
| | RE-Ext(Jie et al., 2022) | 78.6 |
| | M-View◇(Zhang et al., 2022a) | 79.5 |
| | Elastic ♣(Zhang and Moshfeghi, 2022) | 80.3 |
| | MWP-NAS†(Bin et al., 2023) | 79.2 |
| **LLM** | gpt-3.5-turbo†(OpenAI, 2022) | 42.6 |
| | Self-Consistency†(Wang et al., 2022b) | 50.7 |
| | Ours | **81.5** ±0.13 |
| | Ours (Layer-Shared) | **81.1** ±0.23 |

Table 1: Results on MathQA. † means our reproduction. ◇ means we reproduce *M-View* using the standard dataset without data Augmentation (their report). ♣ means Elastic use a different data pre-processing method and operators, so we reproduce their method.

| | Model | Test | 5-fold |
|---|---|---|---|
| **Seq2Seq / Tree** | GroupAttn(2019) | 69.5 | 66.9 |
| | GTS (2019) | 75.6 | 74.3 |
| | G2T(2020) | 77.4 | 75.5 |
| | mBERT(2021) | 75.1 | - |
| | Symbol-Dec(2021) | - | 75.7 |
| | BERTGen(2021) | 76.6 | - |
| | PLM-Gen (2021) | 76.9 | - |
| | H-Reasoner(2021) | 83.9 | 82.2 |
| | BERT-T(2021a) | 84.4 | 82.3 |
| | Rank†(2021) | 85.4 | - |
| | Logic-Dec(2022b) | 83.4 | - |
| | T-Dis(2021b) | 79.1 | 77.2 |
| | Prototype (2021) | 83.2 | - |
| | Textual-CL (2022a) | 85.0 | 82.6† |
| | Ana-CL (2022) | 85.6 | 83.2† |
| **Seq2Exp** | E-Pointer†(2020) | 78.7 | 76.5 |
| | DAG (2021) | 77.5 | 75.1 |
| | M-Tree(2022a) | 82.5 | 80.8† |
| | RE-Ext(2022) | 85.4 | 83.3 |
| | M-View◇(2022a) | 85.6 | 83.1 |
| | Elastic†(2022) | 84.8 | 82.9 |
| | MWP-NAS(2023) | 84.4 | |
| **LLM** | gpt-3.5-turbo†(2022) | 54.8 | - |
| | Self-Consistency†(2022b) | 66.1 | - |
| | Ours | **86.2** ±0.30 | **84.1** ±0.65 |
| | Ours (Layer-Shared) | **85.6** ±0.25 | **83.4** ±0.38 |

Table 2: Testing and five-fold Acc. on Math23k.

## 3.4 Training and Inference

During training, we calculate the loss after employing the bipartite match for each layer. Besides, we also adopt the teacher forcing (Williams and Zipser, 1989) by using golden expressions for the next layer (Equation 5, 6). During inference, each layer predicts $K$ expressions in parallel. Those expressions whose predicted operator is $None$ are filtered out before proceeding to the next layer generation. The entire process is finished when all $K$ expressions are predicted to be $None$.

## 4 Experiments

**Math Word Problem Task** We first evaluate our expression tree decoding strategy on the Math Word Problem task (MWP). MWP represents a challenging mathematical reasoning task where the model is required to comprehend the semantics of a given problem and generate a complete equation and the solution. Encompassing a spectrum of topics, including engineering, arithmetic, geometry, and commerce, MWP effectively evaluates the model's prowess in semantic understanding as well as mathematical symbol generation. We use three standard MWP datasets[1] across two languages:

MathQA (Amini et al., 2019), Math23K (Wang et al., 2017), and MAWPS (Koncel-Kedziorski et al., 2016). We follow (Jie et al., 2022; Zhang et al., 2022a) to preprocess datasets. The statistics of datasets are reported in Appendix A.3.

**Baselines** We compare our method with three types of baselines: (1) **Seq2Seq/Tree**: PLM-Gen (Lan et al., 2021), Rank (Shen et al., 2021), Symbol-Dec (Qin et al., 2021), H-Reasoner (Yu et al., 2021), Logic-Dec (Yang et al., 2022b), Prototype (Li et al., 2021), T-Dis (Liang et al., 2021b), Textual-CL (Shen et al., 2022a), Ana-CL (Liang et al., 2022) and several representative methods. (2) **Seq2Exp**: **E-Pointer** (Kim et al., 2020), DAG (Cao et al., 2021), RE-Ext (Jie et al., 2022), M-View (Zhang et al., 2022a), ELASTIC(Zhang and Moshfeghi, 2022), M-Tree (Wang et al., 2022a) and MWP-NAS (Bin et al., 2023). Besides, we also compare with gpt-3.5-turbo (OpenAI, 2022) and Self-Consistency (Wang et al., 2022b) prompted by one demonstration through the OpenAI API. More details are listed in Appendix A.1.

**Training Details** Following most previous works (Zhang et al., 2022a; Jie et al., 2022), we report the average accuracy (five random seeds) with

---

[1]The criteria for the selection of the dataset: 1. Dataset size. 2. Datasize label includes not just answers but also complete

equations. 3. Extensively employed in prior research.

| | Model | 5-fold Acc. |
|---|---|---|
| Seq2Seq / Tree | GroupAttn(2019) | 76.1 |
| | GTS (2019) | 82.6 |
| | G2T(2020) | 85.6 |
| | Rank(2021) | 84.0 |
| | BERTGen(2021) | 86.9 |
| | PLM-Gen (2021) | 88.4 |
| | PLM-GTS (2021a) | 88.5 |
| | PLM-G2T (2021a) | 88.7 |
| | H-Reasoner(2021) | 89.8 |
| | T-Dis(2021b) | 84.2 |
| | Prototype† (2021) | 89.6 |
| | Textual-CL† (2022a) | 91.3 |
| | Ana-CL†(2022) | 91.8 |
| Seq2Exp | E-Pointer (2020) | 83.4 |
| | M-Tree(2022a) | 82.0 |
| | RE-Ext(2022) | 92.2 |
| | M-View◇ (2022a) | 92.1 |
| | Elastic ♣(2022) | 91.8 |
| | MWP-NAS(Bin et al., 2023) | 88 |
| LLM | gpt-3.5-turbo†(2022) | 91.5 |
| | Self-Consistency†(2022b) | **92.5** |
| | Ours | 92.3 $\pm$ 0.41 |
| | Ours (Layer-Shared) | 92.2 $\pm$ 0.28 |

Table 3: Five-fold cross-validation results on MAWPS.

| Math23K | Seq2Exp | Our | Seq2Seq | Seq2Tree |
|---|---|---|---|---|
| Avg Step | 2.4 | **1.92** | 7.01 | 5.62 |
| Std Step | 1.22 | **0.8** | 3.4 | 2.1 |
| Max Step | 9 | **8** | 27 | 19 |
| **MathQA** | **Seq2Exp** | **Our** | **Seq2Seq** | **Seq2Tree** |
| Avg Step | 4.33 | **3.2** | 16.74 | 9.87 |
| Std Step | 2.26 | **1.6** | 11.03 | 5.51 |
| Max Step | 11 | **8** | 109 | 55 |

Table 4: The statistics for decoding steps between four types of methods.

| Variant | Acc. |
|---|---|
| Bipartite Matching | **81.5** $\pm$ 0.13 |
| Sequence Matching | 78.8 $\pm$ 0.27 |
| Random Matching | 20.1 $\pm$ 1.55 |
| w/ Operand $None$ Loss | 80.7 $\pm$ 0.36 |
| w/o Operator $None$ Loss | 79.6 $\pm$ 0.21 |
| w/o Parallel decoding | 79.9 $\pm$ 0.31 |

Table 5: Ablation on MathQA about bipartite matching.

standard deviation for Math23K and MathQA, and 5-fold cross-validation for Math23K and MAWPS. The test-set accuracy is chosen by the best dev-set accuracy step. Since most of the math problems only require two or three mathematical expressions to be generated in parallel for each step, we set the number of queries $K$ to be 6, which is sufficient to cover all cases. Except for using a standard decoder for layer-wise decoding (**Our**), we also explore an alternate variant (**Our Layer-Shared**), in which parallel decoding is performed at every $N$ transformer layer, but each decoding step shares the parameter of these $N$ layers. This model is efficient with fewer parameters. Model details are reported in Figure A1.

### 4.1 Results

As shown in Table 1, 2 and 3, our expression tree decoding strategy achieves SoTA performance on two large datasets, especially on the most difficult MathQA with +1.2% gains. Similarly, we gain +0.6% (test) and +0.8% (5-fold) improvements on Math23K and comparable performance on the MAWPS. Moreover, we also notice that our performance still substantially outperforms LLMs in the case of complex mathematical equation generation (MathQA: +30.8% and Math23K: +20.1%). Furthermore, our variant model (Our w/ Layer-Shared) has also demonstrated comparable performance

(+0.8% on MathQA), with fewer parameters.

From the view of three types of methods, our method is more stable and effective, with +1.9% and +1.2% gains against the best Seq2Tree baseline (Ana-CL) and best Seq2Exp baseline (Elastic) on MathQA. The Seq2Exp focuses on generation at the sequence at the expression-level, while Seq2Tree absorbs the feature of a tree structure. In contrast, our expression tree decoding strategy, more than generating the expressions, also integrates the tree structure into the parallel decoding process, performing well for complex equations.

In addition to the accuracy comparison, we also analyze the difference in the number of decoding steps. We report the average decoding steps, step standard deviation, and maximum steps for the four types of methods (Seq2Seq: mBERT, Seq2Tree: Ana-CL, Seq2Exp: RE-Ext, and Expression-tree: Ours) in Table 4. We observe that the decoding steps of the token-level generation methods (e.g., Seq2Seq and Seq2Tree) are significantly higher than those of Expression-level methods (about four to five times). Compared to other methods, our parallel decoding method requires fewer decoding steps, especially on the more complex MathQA. We offer some intuitive examples in Figure A2. These results suggest that our parallel strategy **not only offers superior accuracy but also reduces the number of decoding steps.**

| Equation Type | Diagram | Example |
|---|---|---|
| Single | Exp 1 | $a \div b$ |
| Expression Chain | Exp 1 → Exp 2 → Exp 3 | $(a + b) \times c - d$ |
| Expression Tree | Exp 1, Exp 2, Exp 3 → Exp 4 → Exp 5 | $(a + b) \times (c - d) - e \div f$ |

| Model | Single | Exp Chain | Exp Tree | Overall |
|---|---|---|---|---|
| M-View | 79.7 | 82.7 | 65.7 | 79.5 |
| RE-Ext | 77.4 | 82.5 | 64.7 | 78.6 |
| M-Tree | 75.2 | 80.2 | 64.5 | 76.5 |
| Ana-CL | 79.9 | **82.8** | 67.8 | 79.6 |
| Ours | **80.1** | 82.0 | **75.2** | **81.5** |

Table 6: We categorize the structures of equations into three types: Single, Chain and Tree, and evaluate the performance of five methods on three structures.

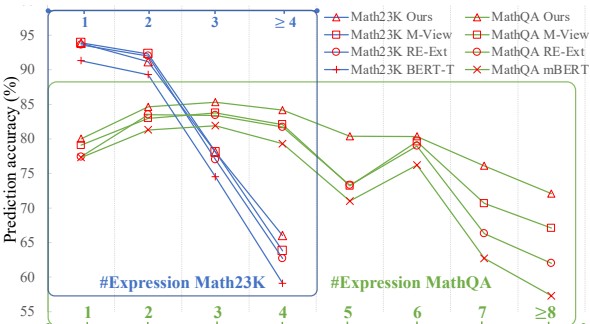

Figure 3: Performance on the sample with the different number of expressions.

## 4.2 Ablations

Bipartite matching is essential for our parallel strategy. Therefore, we study how it works: I. **Sequence Matching**. Firstly, we ablate bipartite matching and instead use a simple matching strategy for multiple expressions: sequence matching. It means we align the first expression predicted by the first query with the first label, and then the second predicted expression aligns with the second label, and so on. II. **Random Matching**. Then we random match the predicted expressions with the labels. III. **Operand None Loss**. As illustrated in Equation 10, for these labels padded with the $None$ category, we only compute the loss for the operator. At this point, we add two operands' loss between $None$ to analyze its effect. IV. **Operator None Loss**. We remove the operator loss for the $None$ category. V. **Parallel Decoding**. Lastly, we remove the whole parallel decoding, i.e., adopt only one query per layer. We provide a detailed visualization for these designs in Figure A3.

As shown in Table 5, when we replace with sequence matching, there is a notable degradation in accuracy (-2.7%). In this case, the performance is similar to the Seq2Exp (RE-Ext:78.6% vs Ablation: 78.8%). It undermines the advantages of expression tree decoding since aligning process still introduces manually-annotated order. Secondly, we find random matching may lead to a training collapse. Then we observe disregarding $None$ operator loss or adding the $None$ operands loss, both having a negative impact (-0.9%, -0.8%). In the last ablation experiments, our performance drops from 81.5% to 79.9% (-1.6%) when we remove the parallel decoding from our system. More comparisons can be found in Appendix A.4.

## 4.3 Analysis

We investigate the efficacy of the expression tree decoding strategy in scenarios involving complex equation generation. We conduct the analysis along two dimensions: the structure of the equation and the length of the equation. Besides, we also analyze the impact of different query numbers on parallel decoding performance.

**Equation Structure** In Table 6, we categorize the structures of equations into three types: (1) Single expression, where the entire equation consists of only one expression; (2) Expression chain, where the equation is comprised of multiple expressions forming a chain; (3) Expression Tree, which involves complex tree structures composed of multiple expressions. We evaluate the model's accuracy on three types of equation structures.

As shown in Table 6, our expression tree decoding strategy gains comparable performance to other baselines in the first two evaluations (Single Expression and Expression Chain). In contrast, in the Expression Tree evaluation, most of these instances in this type involve sophisticated equation structures and complex solving processes. Our method significantly outperforms the other baselines ($\geq +7.4\%$). Specifically, in comparison to the seq2tree approach, we achieve a +7.4% improvement (Our:75.2% vs Ana-CL:67.8%), and gain a more substantial advantage (+9.5%) relative to the seq2exp method. Under this case, our method outperforms seq2tree, and seq2tree in turn outperforms seq2exp. This clearly demonstrates that introducing the structural feature of equations indeed contributes to the capable of handling equations with complex structures.

**Equation Length** An equation comprises multiple mathematical expressions, with each expres-

sion representing a reasoning step. Complex equations usually contain more expressions. Therefore, we evaluate the performance on the instance with different numbers of expressions. In Figure 3, as the number of expressions increases, the equation becomes more complex and the performance decreases rapidly. However, our method consistently maintains high accuracy ($\geq$70% on MathQA) across all cases, especially on complex cases. Compared with baselines, our advantage increases from +1.0% (#2) to +6.4% (#5). For the equation with the longest expressions ($\geq$#8), our strategy maintains an improvement by nearly +6%, showing expression tree decoding strategy is more stable for complex equations.

**Query Number** We further analyze the impact of the number of queries on parallel decoding performance. The number of queries is set from 1 to 30. As shown in Table A2, as the number of queries increases, the performance initially increases notably and then decreases. Specifically, when there is only one query per layer (# query = 1), the parallel decoding strategy is removed. Conversely, when we adopt too many queries (# query >= 10), the performance of parallel decoding drops rapidly. We speculate that this might be because most of the queries are redundant and are matched to the "None" label under this case. Too many queries may lead to instability in training. Apart from too many or too few queries, the performance gains attributed to parallel decoding remain both stable and pronounced. For instance, as the number of queries fluctuates between 4 and 8, the improvement consistently remains within the range of +1.2% to +1.5%. It suggests that although the number of queries is a hyperparameter, it does not need to be carefully tuned.

### 4.4 Case Study and Visualization

We explore the role of learnable queries in the expression tree decoding process. We first calculate the similarity between query vectors and problem representations for each layer and then visualize the results in Figure A4. As shown in the first case, the sixth query is activated twice through two steps, thus performing two division operations. In the second case, the first and second queries generate two valid expressions $(14 + 6, 4 + 6)$ in parallel in the first layer, and then the last query in the second layer outputs a division operation $(Exp_1 + Exp_2)$ using two results from the first layer. These exam-

ples illustrate that our method is highly flexible and can adaptively predict expressions in parallel or sequentially based on the context, thereby composing an expression tree.

## 5 Conclusion

We devise an expression tree decoding strategy for generating mathematical equations layer by layer. Each layer produces multiple mathematical expressions in parallel which are non-dependent and order-free, achieving flexible decoding. During the training, we employ a bipartite matching algorithm to align the multiple generated expressions with the golden labels and compute the parallel prediction loss under the optimal matching scheme. Extensive experiments demonstrate that our expression tree decoding strategy can effectively absorb the structural features of equations and enhance the capacity for generating complex equations for math reasoning.

## Limitations

Firstly, when faced with a mathematical problem that requires an extensive number of solving steps, we have to increase the number of decoder layers. It consequently leads to an increase in the model parameters. This is due to our layer-wise decoding strategy, where more complex equations require additional decoding layers. To address this, we have designed a variant model with shared parameters (Layer-Shared in Figure A1), which achieves comparable results without modifying layer number.

Secondly, some hyperparameters (e.g., the number of queries and layer), need to be manually adjusted according to the dataset. In the future, we will explore how to utilize our query-based, layer-wise expression tree decoding strategy to address a broader range of structured generation tasks.

## Acknowledgments

This work is supported by the Fundamental Research Funds for the Central Universities (No. 226-2023-00060), Key Research and Development Program of Zhejiang Province (No. 2021C01013), National Key Research and Development Project of China (No. 2018AAA0101900), Joint Project DH-2022ZY0013 from Donghai Lab, and MOE Engineering Research Center of Digital Library.

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

# A Appendix

## A.1 Baselines

In recent years, the MWP task has garnered widespread attention (Zhou et al., 2023; Xiong et al., 2022; Lan et al., 2022; Yang et al., 2022a; Cobbe et al., 2021; Tsai et al., 2021; Huang et al., 2021). We divide the prior baselines into two categories: Seq2Seq/Tree and Seq2Exp. In Seq2Seq/Tree, Li et al. (2019) (**GroupAttn**) applied a multi-head attention approach using a seq2seq model. Xie and Sun (2019) proposed a seq2tree generation (**GTS**). Zhang et al. (2020) (**G2T**) introduced a graph encoder. Patel et al. (2021b); Liang et al. (2021a) added a PLMs encoder to GTS and G2T (**PLM-GTS, BERT-T**). Tan et al. (2021) proposed a multilingual model (**mBERT**). Lan et al. (2021) utilized Transformer for generation (**BERTGen**). Shen et al. (2021) proposed a multi-task method (**Rank**). Qin et al. (2021) introduced a neural symbolic method (**Symbol-Dec**). Yu et al. (2021) extracted hierarchical features for encoder (**H-Reasoner**). Yang et al. (2022b) designed logical rules to guide decoding (**Logic-Dec**). Li et al. (2021) proposed a prototype learning (**Prototype**). Liang et al. (2021b) adopted a teacher model for discrimination (**T-Dis**). Shen et al. (2022a) distinguished examples with similar semantics but different logics (**Textual-CL**). Liang et al. (2022) adopted an analogy identification to improve the generalization (**Ana-CL**).

In Seq2Exp, Cao et al. (2021) used a bottom-up DAG construction method (**DAG**). Jie et al. (2022) introduced a relation extraction method (**RE-Ext**). Wang et al. (2022a) treated MWP as tagging annotation by M-Tree coding (**M-Tree**). Bin et al. (2023) introduced a unified tree structure using a non-autoregressive model (**MWP-NAS**). Zhang et al. (2022a) aligned the representation of different traversal order for consistency (**M-View**). ELASTIC (Zhang and Moshfeghi, 2022) designs a computer synthesis process to handle numerical reasoning. We also compare our results with the gpt-3.5-turbo in the few-shot setting. We design a prompt consisting of both directive instructions and a demonstration to guide gpt-3.5 step-by-step reasoning.

## A.2 Training Details

Following most previous works (Zhang et al., 2022a; Jie et al., 2022), we adopt Roberta-base and Chinese-BERT as encoder from HuggingFace

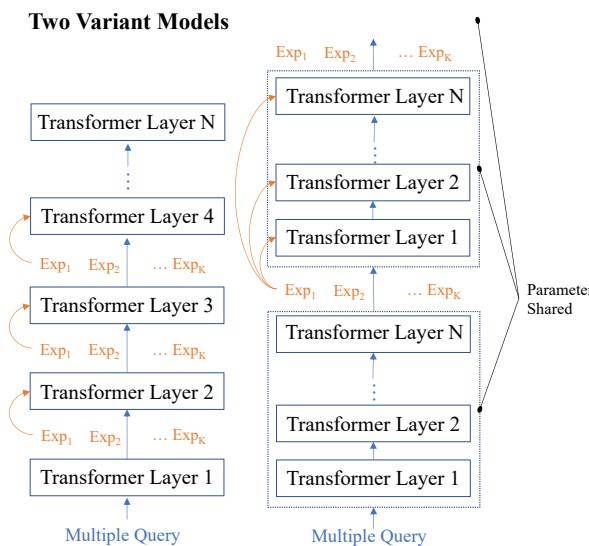

Figure A1: Except for using a standard decoder for layer-wise decoding (Left), we also explore an alternate model (Right), in which parallel decoding is performed at every $N$ transformer layer, but each decoding step shares the parameter of these $N$ layers. The left model is more accurate, and the right has fewer parameters.

(Wolf et al., 2020) for multilingual datasets. We consider five mathematical operators, containing *Addition, Subtraction, Multiplication, Division, Exponentiation*, and various constants ($\{\pi, 1, 0, \cdots\}$) as previously. Our query decoder is a transformer decoder with multiple layers, each having 768 hidden units. In our experiments, we perform parallel decoding once at each transformer layer. We use an AdamW optimizer with a 5e-5 learning rate, batch size of 32 for MathQA and 26 for Math23K. We set the maximum layer number as 8. The other parameters are set as previous works (Zhang et al., 2022a; Jie et al., 2022). All experiments were set up on an NVIDIA RTX A6000.

## A.3 MWP Dataset Statistics

The statistics of the dataset are shown in Table A1.

| Dataset | #Train/#Valid/#Test | #Avg. Token | #Avg. Exp | #Max. Exp |
|---|---|---|---|---|
| MathQA | 16191 /2415/1606 | 39.6 | 4.17 | 12 |
| Math23K | 21162/1000/1000 | 26.6 | 2.26 | 20 |
| MAWPS | 1589/199/199 | 30.3 | 1.42 | 7 |

Table A1: Statistics for three standard datasets.

| # Query | 1 | 2 | 4 | 5 | 6 | 7 | 8 | 10 | 15 | 20 | 30 |
|---|---|---|---|---|---|---|---|---|---|---|---|
| **Acc.(MathQA) %** | 79.9 | 80.6 | 81.4 | 81.4 | 81.5 | 81.3 | 81.1 | 80.8 | 80.2 | 79.8 | 79.1 |
| **Compared to #1** | 0 | +0.7 | +1.5 | +1.5 | +1.6 | +1.4 | +1.2 | +0.9 | +0.3 | -0.1 | -0.8 |
| **Acc.(Math23K) %** | 85.2 | 85.8 | 86.2 | 86.0 | 86.2 | 85.9 | 85.6 | 85.2 | 84.8 | 83.5 | 83.3 |
| **Compared to #1** | 0 | +0.6 | +1 | +0.8 | +1 | +0.7 | +0.4 | 0 | -0.4 | -1.7 | -1.9 |

Table A2: The impact of query number on our parallel decoding performance.

| Method | MathQA | Math23K |
|---|---|---|
| w/ parallel decoding | 81.5 | 86.2 |
| w/o parallel decoding | 79.9 | 85.2 |
| E-pointer | 73.5 | 78.7 |
| M-View | 79.5 | 85.6 |
| RE-Ext | 78.6 | 85.4 |
| Elastic | 80.3 | 84.8 |

Table A3: The ablation study on parallel decoding.

## A.4 Ablating on Parallel Decoding

Parallel decoding is the key to constructing expression trees. We provide a more detailed comparison of our parallel decoding strategy. The detailed results are as shown in Table A3. When we ablate parallel decoding from our framework, i.e., adopt only one query per layer, our performance drops from 81.5% to 79.9% (-1.6%) on MathQA. A similar trend is seen on Math23K (-1.0%). Besides, without the parallel decoding strategy, our performance is similar to the Seq2Exp baselines (e.g., E-pointer, M-View, RE-Ext, Elastic, etc.), which generate one expression at each step. Compared to them, parallel decoding brings noticeable and consistent improvements (E-pointer: +8%, M-View: +2%, RE-Ext: +2.9%, Elastic: +1.2%).

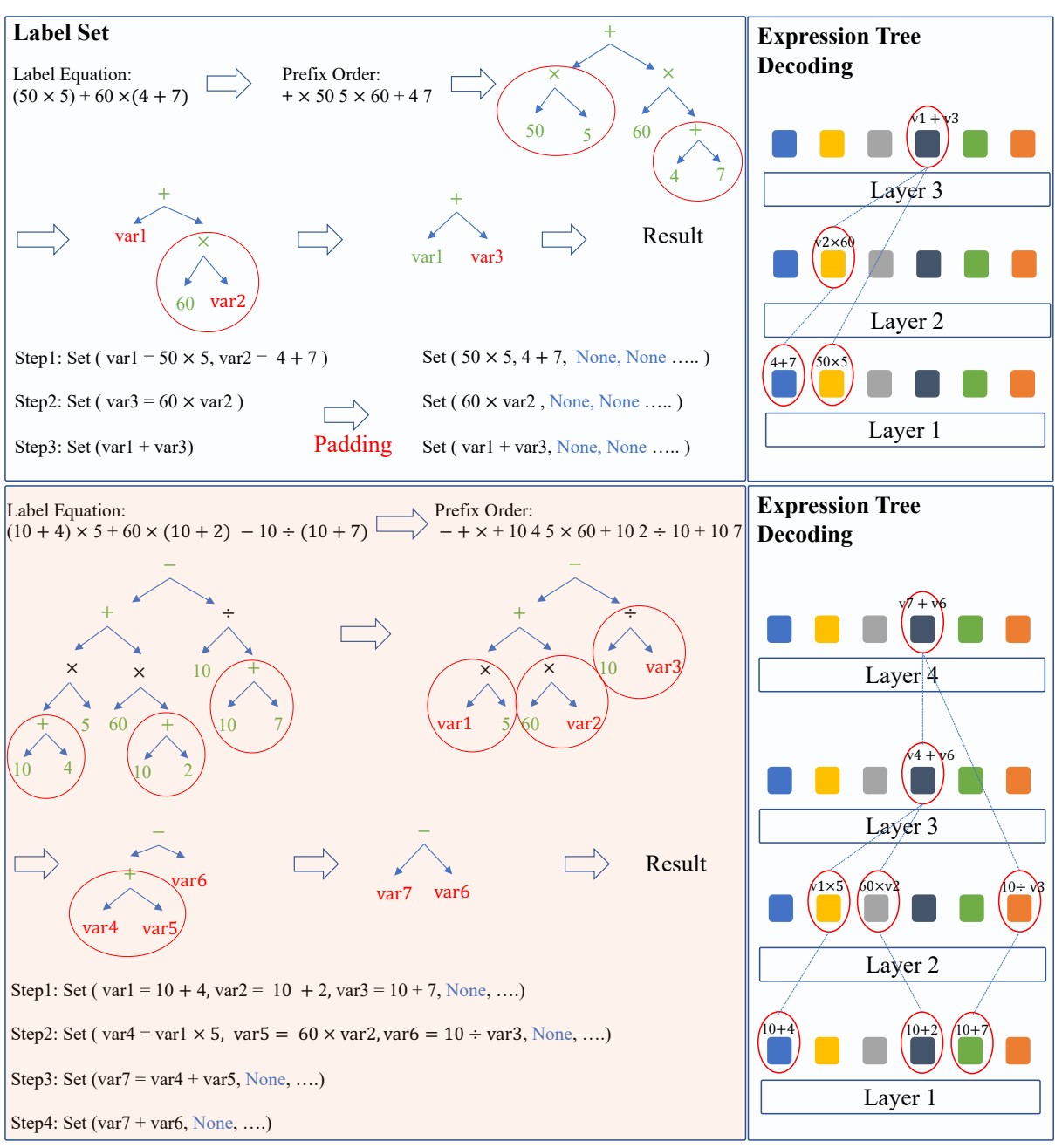

Figure A2: Two cases for Label Set and Expression Tree decoding processes.

**Bipartite Matching Process**

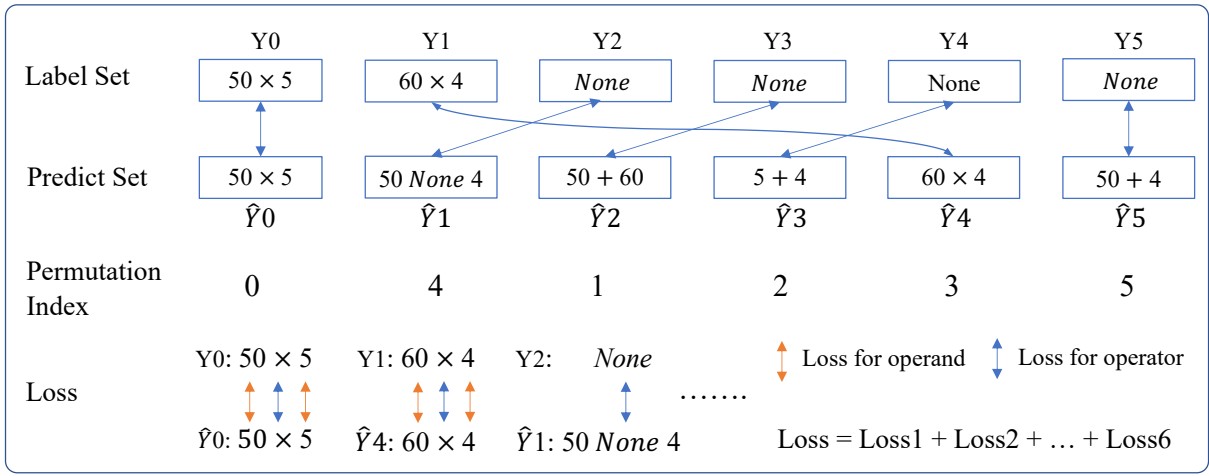

**Several Ablation Design**

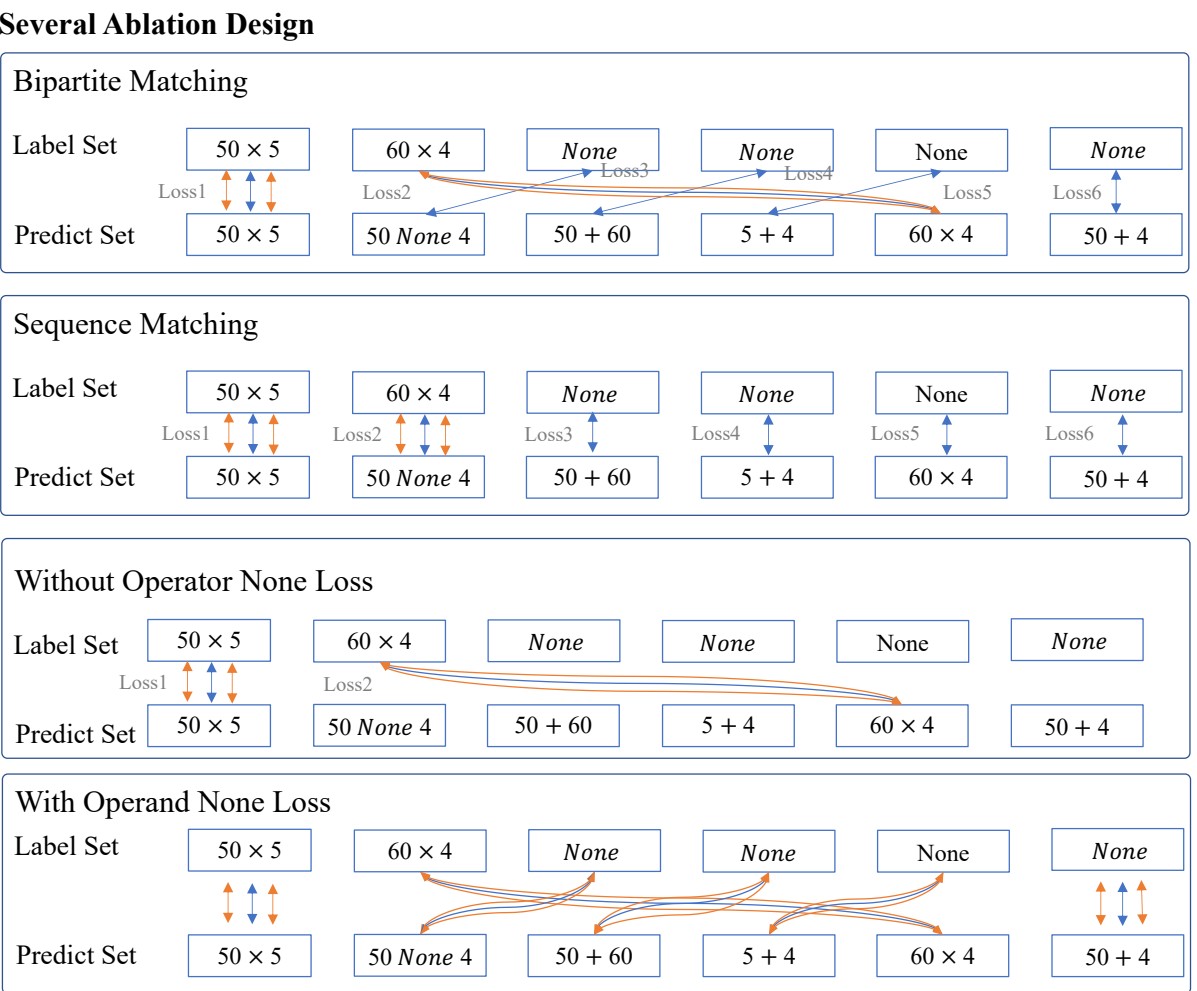

Figure A3: Up: The process of the Bipartite Matching. Down: Several ablation designs.

**Problem**: if a truck is traveling at a constant rate of 180 kilometers per hour , how many hours will it take the truck to travel a distance of 600 meters ? ( 1 kilometer = 1000 meters )

**Label**: 600 / 1000 / 180       **Our Prediction**: 600 / 1000 / 180

**Layer 1: Exp1= 600 ÷ 1000 by # query 6**

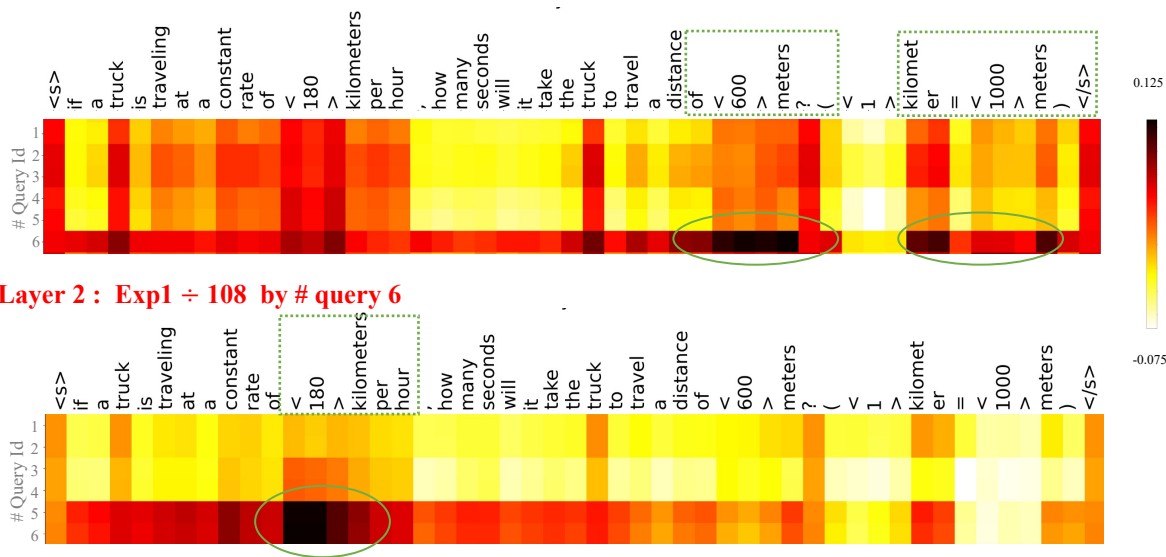

**Layer 2 : Exp1 ÷ 108 by # query 6**

**Problem**: maxwell leaves his home and walks toward brad ' s house . one hour later , brad leaves his home and runs toward maxwell ' s house . if the distance between their homes is 14 kilometers , maxwell ' s walking speed is 4 km / h , and brad ' s running speed is 6 km / h . what is the total time it takes maxwell before he meets up with brad ?

**Label** : (14+6) / (4+6)       **Our Parallel Prediction**: ( 14 + 6 ) ÷ ( 4 + 6 )

**Layer 1: Exp1 = 14 + 6 by # query 1       Exp2 = 4 + 6 by # query 2**

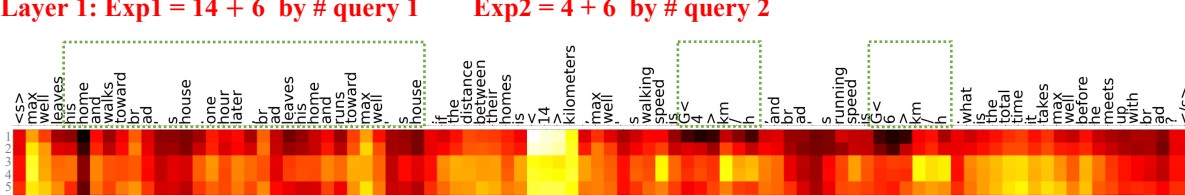

**Layer 2: Exp1 ÷ Exp2 by # query 6**

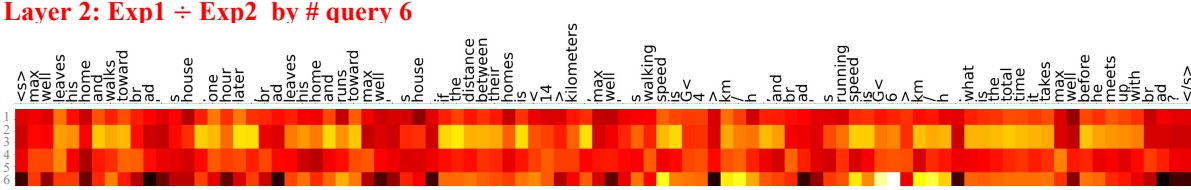

Figure A4: We visualize the expression tree decoding process at each layer. We calculate the cosine similarity between query vectors and problem representations for each layer. In the first case, the prediction expressions are output by a query. In the second case, the first and second queries are activated for two expressions in parallel.