# OpenReview forum: "An Expression Tree Decoding Strategy for Mathematical Equation Generation"
_EMNLP/2023/Conference — EMNLP 2023 Main_

### Official Review · Reviewer_1r8v · 2023-07-28

**Soundness:** 4

**Excitement:**

3: Ambivalent: It has merits (e.g., it reports state-of-the-art results, the idea is nice), but there are key weaknesses (e.g., it describes incremental work), and it can significantly benefit from another round of revision. However, I won't object to accepting it if my co-reviewers champion it.

**Missing References:**

- The paper would benefit from establishing a better relation to classical methods in parsing or generation.
The different tree construction scenarios investigated in the paper share striking similarities with parsing methods : top down, shift reduce, CKY... Relating the proposal with well known methods would add some substantial value to the paper.

**Paper Topic And Main Contributions:**

The authors describe a mathematical expression generator from natural language.
The context is to translate a mathematical problem expressed in natural language to an arithmetic expression

In the current research context, the solution builds upon transformers
in a seq2expression framework: that is given an encoding of the natural language, generate the expression.
Here the authors focus on generating the expression tree rather than generating directly the expression tokens sequentially.

In this case, several solutions are possible : prefix, infix, postfix encoding of the expression tree.
The authors observe that these orderings involve a large number of steps.

They introduce a new method that attempts to minimize the number of steps for generating the expression structure. The method builds the expression tree iteratively in bottom up order.
At each step the system issues K queries to the problem representation, each query generates a math expression
each of these valid expressions are then added to the problem embedding and feed the next step.
The process stops when no valid expression is generated at some step

The authors also describe a method for constructing a loss in this setup relying on an alignment between the reference solution and the current state of the learner relying on bipartite matching method

The authors then provide experimental results on the Math Word Problem task (MWP) for 3 datasets where they get sota results. An analysis is provided where the relevance of the bipartite matching component is first highlighted.
Then the experiments show that their method is well suited for long equations and equations with non trivial tree structure









**Questions For The Authors:**

An analysis of the number of steps required to generate the expressions would be interesting too.
In the examples given, your method takes slightly less steps than the postfix method. Providing an analysis of the expected number of steps for an expression between the different methods would be a nice add-on

**Reasons To Accept:**

- The authors provide an original method for generating arithmetic expressions from problems stated in natural language
- The results are good and the analysis highlights where the systems improves over the sota

**Reasons To Reject:**

- The method is dependent on a ad-hoc constant:
there's is an hyper parameter constant K throughout the paper (the number of queries issued at each step) that is ad-hoc.
- The paper would benefit from establishing a better relation to classical methods in parsing or generation.
The different tree construction scenarios investigated in the paper share striking similarities with parsing methods : top down, shift reduce, CKY... Relating the proposal with well known methods would add some substantial value to the paper.

**Reproducibility:**

4: Could mostly reproduce the results, but there may be some variation because of sample variance or minor variations in their interpretation of the protocol or method.

**Reviewer Confidence:**

3: Pretty sure, but there's a chance I missed something. Although I have a good feel for this area in general, I did not carefully check the paper's details, e.g., the math, experimental design, or novelty.

---

> ### Author Rebuttal · Authors · 2023-08-27
>
> #### The response to Reviewer 1r8v
>
> Thanks for your valuable comments. Below are our responses to your concerns:
>
> > ***Reject1: The number of queries is a hyperparameter***.
>
> Thank you for pointing out this limitation. We also discussed it in the Limitation Section. We found that, although the number of queries needs to be predefined, it doesn't require meticulous design:
>
>
>
> + **Performance is stable with different number of queries**. We analyze the impact of the query number. We set the number of queries (K) from 1 to 30. We noticed that the performance is stable and satisfactory (+1.5\%, +1.5\%, +1.6\%, +1.4\%, +1.2\%) when K is set between 4 and 8. This suggests that although the number of queries is a hyperparameter, it does not need to be carefully tuned. A moderate K can lead to a decent promotion.
>
>
> + **The number of queries is easy to estimate**.
> Contrary to being ad-hoc, the number of queries $K$ is quite intuitive to estimate. Our experimental results clearly demonstrate that extreme values (either too many or too few queries) are suboptimal. In real-world scenarios, most mathematical equations comprise only a handful of expressions that need parallel output. As a result, setting $K$ to a small integer (like within the range of 4 to 8) is not only empirical but also logical.
>
> The detailed results for different query numbers:
>
> |   # query         | 1   |   2   | 4   | 5   | 6   | 7   | 8   | 10  | 15  | 20  | 30  |
> |:----------------:|:---:|:---:|:---:|:---:|:---:|:---:|:---:|:---:|:---:|:---:|:---:|
> | Acc.(MathQA) %    | 79.9| 80.6| 81.4| 81.4| **81.5**| 81.3| 81.1| 80.8| 80.2| 79.8| 79.1|
> | Comp to #1       | 0   | +0.7| +1.5| +1.5| **+1.6**| +1.4| +1.2| +0.9| +0.3| -0.1| -0.8|
>
>
> |      # query        |  1  |  2  |  4  |  5  |  6  |  7  |  8  | 10  | 15  | 20  | 30  |
> |:----------------:|:---:|:---:|:---:|:---:|:---:|:---:|:---:|:---:|:---:|:---:|:---:|
> | Acc.( Math23K)   | 85.2| 85.8| **86.2**| 86.0| **86.2**| 85.9| 85.6| 85.2| 84.8| 83.5| 83.3|
> | Comp to #1       |  0  | +0.6| **+1**  | +0.8| **+1**| +0.7| +0.4|  0  | -0.4| -1.7| -1.9|
>
>
>
> > ***Reject2: Establishing the association between different equation generation methods and parsing***
>
> Thank you for your insightful suggestion.
>
> It is quite intriguing to consider different equation generation methods from the perspective of parsing tasks. The Seq2Tree equation generation can be viewed as a shift-reduce strategy. Our approach also shares some similarities with the classic CKY algorithm. Firstly, both methods construct tree structures in a bottom-up manner, layer by layer. Secondly, while CKY employs dynamic programming to seek the optimal substructure, we use a bipartite matching (Hungarian algorithm) to find an optimal matching relation. We will add discussions about these parsing algorithms in our revised version.
>
>
> > ***Question: Analyse the number of decoding steps between different methods***
>
>
> We calculate the average decoding steps, step standard deviation, and maximum decoding steps for the four types of methods (Seq2Seq: mBERT, Seq2Tree: Ana-CL, Seq2Exp: RE-Ext, and Expression-tree: Ours). We observed the following phenomena:
>
> + Seq2Seq and Seq2Tree are token-level generation methods, so their decoding steps are significantly higher than those of Expression-level methods (about four to five times).
>
>
> + Compared to Seq2Exp, our parallel decoding method requires fewer decoding steps (Math23K: -0.48, MathQA: -1.13) and Std, especially on the more complex MathQA. We offer some intuitive examples in Figure A2. These results suggest that our parallel decoding strategy not only offers superior accuracy but also reduces the number of decoding steps.
>
>
> |    Math23K      | Seq2Exp | Our | Seq2Seq | Seq2Tree |
> |----------|---------|-----|---------|----------|
> | Avg Step | 2.4     | **1.92**| 7.01    | 5.62     |
> | Std Step | 1.22    | **0.8** | 3.4     | 2.1      |
> | Max Step | 9       | **8**   | 27      | 19       |
>
> |  MathQA        | Seq2Exp | Our  | Seq2Seq | Seq2Tree |
> |----------|---------|------|---------|----------|
> | Avg Step | 4.33    | **3.2**  | 16.74   | 9.87     |
> | Std Step | 2.26    | **1.6** | 11.03   | 5.51     |
> | Max Step | 11      | **8**   | 109     | 55       |
>
>
> -----
> Thanks again for your valuable reviews above. If you have any further questions, please do not hesitate to let me know. I am more than happy to discuss with you (if the system allows).

---

### Official Review · Reviewer_Apu7 · 2023-08-03

**Soundness:** 3

**Excitement:**

4: Strong: This paper deepens the understanding of some phenomenon or lowers the barriers to an existing research direction.

**Missing References:**

None

**Paper Topic And Main Contributions:**

This paper proposes a method for math word problems. The proposed method partitions nodes in an expression tree by their layer (from left to root), and iteratively combines sub-expressions and variables to construct larger sub-expression. At each step, it generates all sub-expressions in the same layer in parallel, using a method similar to DETR (a multi-object detection framework).

**Questions For The Authors:**

My first two questions are listed in the Reasons to Reject.
C. The ablation study only ablates the bipartite matching method, is it possible to ablate other parts of the mode like parallel decoding?


**Reasons To Accept:**

1. From my understanding, this paper combines the method from [1] and [2], where [1] constructs the tree in a bottom-up layer-wise fashion and [2] proposes a method to extract multiple objects in each layer in parallel. This combination is interesting since [2] is a paper on object detection tasks in CV.
2. The experiment results are promising.

[1] Yixuan Cao, Feng Hong, Hongwei Li, and Ping Luo. 2021. A bottom-up dag structure extraction model for math word problems. In AAAI.
[2] Nicolas Carion, Francisco Massa, Gabriel Synnaeve, Nicolas Usunier, Alexander Kirillov, and Sergey Zagoruyko. 2020. End-to-end object detection with transformers. In ECCV.

**Reasons To Reject:**

A. While from my understanding, [1] and [2] are the most important references for this paper, the authors have introduced them very briefly.
B. The authors claim that ``To the best of our knowledge, this is the first effort to parallel decode expressions and form an expression tree for MWP.'' But [1] is a parallel decoding method, which has a similar decoding process to this paper.


**Reproducibility:**

4: Could mostly reproduce the results, but there may be some variation because of sample variance or minor variations in their interpretation of the protocol or method.

**Reviewer Confidence:**

4: Quite sure. I tried to check the important points carefully. It's unlikely, though conceivable, that I missed something that should affect my ratings.

**Typos Grammar Style And Presentation Improvements:**

     a. In line 256, "K query embeddings" should be "K query hidden vectors" to distinguish the embedding matrix and computed results.
     b. In line 272, "We totally predict K independent expressions from K query at this layer."

---

> ### Author Rebuttal · Authors · 2023-08-27
>
> #### Reviewer Apu7
>
> Thanks for your valuable comments. Below are our responses to your concerns:
>
> > ***Question A: Introducing Seq2DAG and DETR is too brief***
>
> Thank you for highlighting the importance of [1] and [2] in our work. In our revised paper, we will discuss both papers in detail.
>
> Seq2DAG [1] and DETR [2] are two profoundly significant works, which inspired us in many aspects.
> To be specific, Seq2DAG's framework provided us with insights on the parallel extraction of multiple equations, and DETR's query-based detection mechanism served as a foundation for our method to predict sub-expression nodes for complex equations.
>
> In the revised version, we will discuss more about [1-2] and delve into their methodologies, their main contributions, and how our work draws from these two pioneering studies.
>
>
> > ***Question B: Seq2DAG is also a parallel decoding method***
>
> Sorry for the ambiguity in our statement. We will revise this statement and discuss their work further. Seq2DAG is the first to perform a bottom-up structure extraction, greatly inspiring the MWP community. There are some comparisons between Seq2DAG and our method:
>
> + **Different Focus on Parallel Decoding:** Both methods involve parallel decoding, but they have different concerns.
> Seq2DAG extracts multiple equations in parallel, i.e., an annotated equation set (e.g., x+y=3, y-2=4), whereas our method considers how to predict multiple sub-expressions in parallel for a complex equation at each step. We need to craft a label set for the parallel expressions at each layer ourselves (Section 3.3). For instance in Figure A2, the equation (10 + 4) × 5 + 60 × (10 + 2) − 10 ÷ (10 + 7) is converted into four label sets: layer1= {10+4, 10+2, 10+7}, layer2={v1×5, 60×v2, 10÷v3}, layer3={v4+v5}, layer4={v6+v7}. This process is not mentioned in Seq2DAG.
>
>
> + **Different Methodology:** Seq2DAG is an extraction framework, that enumerates all possible (l, op, r) combinations at each step, and then performs binary classification for each combination. This process is similar to iterative relation extraction [3]. In contrast, our query-based decoding method is a generative framework, that directly generates multiple possible expressions in parallel rather than enumerating all combinations. Then we align predicted expressions with the label set via bipartite matching loss, which is also different from Seq2DAG.
>
> [1] Yixuan Cao, Feng Hong, Hongwei Li, and Ping Luo. 2021. A bottom-up dag structure extraction model for math word problems. In AAAI.
>
> [2] Nicolas Carion, Francisco Massa, Gabriel Synnaeve, Nicolas Usunier, Alexander Kirillov, and Sergey Zagoruyko. 2020. End-to-end object detection with transformers. In ECCV.
>
> [3] Cao, Yixuan and Chen, Dian and Li, Hongwei and Luo, Ping. 2019. Nested relation extraction with iterative neural network. CIKM
>
>
>
> > ***Question C: The ablation study on parallel decoding***
>
> Thank you for your helpful suggestion. We analyze the parallel decoding strategy from two distinct perspectives:
>
> + **Ablating the whole parallel decoding**: Parallel decoding is the key to constructing expression trees. When we ablate parallel decoding from our framework, i.e., adopt only one query per layer, our performance drops from 81.5\% to 79.9\% (-1.6\%) on MathQA. A similar trend is seen on Math23K (-1.0\%). Besides, without the parallel decoding strategy, our performance is similar to the Seq2Exp baselines (e.g., E-pointer, M-View, RE-Ext, Elastic, etc.), which generate one expression at each step. Compared to them, parallel decoding brings noticeable and consistent improvements (E-pointer: +8\%, M-View: +2\%, RE-Ext: +2.9\%, Elastic: +1.2\%). The detailed results are as follows:
> | Method                | MathQA | Math23K |
> |-----------------------|--------|---------|
> | w/ parallel decoding  |  81.5  |   86.2  |
> | w/o parallel decoding |  79.9  |   85.2  |
> | E-pointer             |  73.5  |   78.7  |
> | M-View                |  79.5  |   85.6  |
> | RE-Ext                |  78.6  |   85.4  |
> | Elastic               |  80.3  |   84.8  |
>
>
> + **Changing the number of parallel queries**: We further analyze the impact of the number of queries on parallel decoding performance. The number of queries is set from 1 to 30. The results show as the number of queries increases, the performance initially increases notably and then decreases. Specifically, when there is only one query per layer (#query = 1), the parallel decoding strategy is removed. Conversely, when we adopt too many queries (#query >= 15), the performance of parallel decoding drops rapidly. We speculate that this might be because most of the queries are redundant and are matched to the "None" label under this case. Too many queries may lead to instability in training. Apart from these extreme cases (either too many or too few queries), the performance gains attributed to parallel decoding remain both **stable** and **pronounced**. For instance, as the number of queries fluctuates between 4 and 8, the improvement consistently remains within the range of +1.2\% to +1.5\%.
> |   # query         | 1   |   2   | 4   | 5   | 6   | 7   | 8   | 10  | 15  | 20  | 30  |
> |:----------------:|:---:|:---:|:---:|:---:|:---:|:---:|:---:|:---:|:---:|:---:|:---:|
> | Acc.(MathQA) %    | 79.9| 80.6| 81.4| 81.4| **81.5**| 81.3| 81.1| 80.8| 80.2| 79.8| 79.1|
> | Comp to #1       | 0   | +0.7| +1.5| +1.5| **+1.6**| +1.4| +1.2| +0.9| +0.3| -0.1| -0.8|
> ||||||||||||
> | Acc.( Math23K)   | 85.2| 85.8| **86.2**| 86.0| **86.2**| 85.9| 85.6| 85.2| 84.8| 83.5| 83.3|
> | Comp to #1       |  0  | +0.6| **+1**  | +0.8| **+1**| +0.7| +0.4|  0  | -0.4| -1.7| -1.9|
>
>
>
> > ***Presentation Improvements***
>
> Thanks for pointing out these issues. we will revise them.
>
> ----------------
> Thanks again for your valuable reviews above. If you have any further questions, please do not hesitate to let me know. I am more than happy to discuss with you (if the system allows).

---

### Official Review · Reviewer_GxCv · 2023-08-04

**Soundness:** 4

**Excitement:**

3: Ambivalent: It has merits (e.g., it reports state-of-the-art results, the idea is nice), but there are key weaknesses (e.g., it describes incremental work), and it can significantly benefit from another round of revision. However, I won't object to accepting it if my co-reviewers champion it.

**Paper Topic And Main Contributions:**

The paper is about generating mathematical equations from natural language problem descriptions. The authors propose 'Expression Tree Decoding' as a novel approach that integrates tree structure into existing expression-level (seq2exp) approaches. To this end, they adapt ideas from query-based object detection, especially from [Carion et al., 2020] for math word problems. By using a bipartite matching loss for set prediction, the model is able to predict multiple independent expressions in parallel while dependent expressions are decoded sequentially. Here, a transformer model is used, where each layer performs one decoding step. This approach constructs an expression tree that aligns with the inherent structure of mathematical equations. The authors show their approach to outperform prior baselines on 3 common MWP tasks.

**Questions For The Authors:**

A) The conclusion that 'Expression Tree Decoding' empowers the model to exploit the inherent structure of the equation does not seem to be well proven. Further ablation experiments (e.g. using only 1 query per layer) would be helpful.

B) The section 'Learnable Query' may need clarification, it is not clear to me how these queries (which seem to be embedding vectors?) are learned and used to predict expressions of the given form (l, op, r) .

C) According to Section A.2, your transformer has 8 layers, limiting the number of expressions. Are there examples in the datasets that are too complex for the model and how do you handle them?

D) In the training /evaluation code, you define numerical constants (conf.use_constant in structured_decoding_main_math23k.py) based on the dataset used. I would expect this to affect the evaluation, but it is not mentioned in the paper. In which way do you use these constants and how do the results differ when no constants are used?

E) In Table 3, why do you report accuracy of your models in bold instead of Self-Consistency(2022b), which actually achieves the best result?

**Reasons To Accept:**

- The idea to adapt query-based detection combined with bipartite matching and layer-wise decoding to math word problems seems novel and appropriate.
- The paper is well written and easy to follow.

**Reasons To Reject:**

- The paper mainly employs known strategies to a new problem which might limit the novelty.

**Reproducibility:**

4: Could mostly reproduce the results, but there may be some variation because of sample variance or minor variations in their interpretation of the protocol or method.

**Reviewer Confidence:**

2: Willing to defend my evaluation, but it is fairly likely that I missed some details, didn't understand some central points, or can't be sure about the novelty of the work.

**Typos Grammar Style And Presentation Improvements:**

- l.294:  If all predicted expressions are invalid, i.e., K operators are classified as None, which signifies the equation generation is complete.
- l. 355: we caculate ...
- The example problems provided throughout the paper are very similar (all distance/time/speed related), it would be interesting to see more diverse examples
- The paper has more than 8 pages

---

> ### Author Rebuttal · Authors · 2023-08-27
>
> #### Reviewer GxCv
>
> Thanks for your valuable comments. Below are our responses to your concerns:
>
> > ***Reject 1: The paper employs known strategies to a new problem which might limit the novelty.***
>
> Thank you for taking the time and effort to review our work. We appreciate the opportunity to address the concerns regarding the novelty of our paper:
>
> + **Innovation in Application:**  While the paper might employ known strategies, it creatively applies them to a novel domain, i.e., Equation Generation or Mathematical Word Problems (MWP). Specifically:
>     + We are the first to investigate existing methods from two orthogonal perspectives: token-level or expression-level generation, and sequential order or tree order (Figure 1). From two perspectives, we clearly observe the evolution between the different approaches and identify the weaknesses of the SoTA approach (Expression-Chain generation).
>     + We innovatively incorporate the tree structure of equations into a layer-wise decoding process, generalizing Expression-Chain to a more versatile Expression-Tree generation.
>     + The use of a multi-query parallel strategy and a bipartite matching algorithm in the context of equation generation is a pioneering method (mentioned by Reviewer Apu7). This method not only improves upon previous baselines (+1.2\%, +0.8\% in Table 1, 2) but also demonstrates superior performance in complex equation scenarios (+7.4\% in Table 5, 6).
>
> + **Potential for Broader Impact:** Our paper is more than just simply applying existing strategies to a new problem. Many structured prediction tasks (e.g., Parsing tasks) share similar challenges with equation generation, such as bottom-up or top-down prediction and the problem of optimal sub-structure search. Our approach, especially the multi-query parallel decoding and bipartite matching mechanisms we introduced, may provide insights for these structured prediction tasks. Similarly, as Reviewer 1r8v also mentioned, considering each equation generation method (Seq2Tree, Seq2Exp, Ours) from a Parsing task perspective can bring greater value. We hope our research can bring some inspiration to the community across different domains.
>
> > ***Question A: Ablation experiments about parallel decoding***
>
> Thank you for your helpful suggestion. The crux of Expression-Tree strategy lies in parallel decoding utilizing multiple queries. So we analyze its practical effects from three distinct perspectives:
>
> + **Ablating the whole parallel decoding**: Parallel decoding is the key to constructing expression trees. When we ablate parallel decoding from our framework, i.e., adopt only one query per layer, our performance drops from 81.5\% to 79.9\% (-1.6\%) on MathQA. A similar trend is seen on Math23K (-1.0\%). Besides, without the parallel decoding strategy, our performance is similar to the Seq2Exp baselines (e.g., E-pointer, M-View, RE-Ext, Elastic, etc.), which generate one expression at each step. Compared to them, parallel decoding brings noticeable and consistent improvements (E-pointer: +8\%, M-View: +2\%, RE-Ext: +2.9\%, Elastic: +1.2\%). The detailed results are as follows:
> | Method                | MathQA | Math23K |
> |-----------------------|--------|---------|
> | w/ parallel decoding  |  81.5  |   86.2  |
> | w/o parallel decoding |  79.9  |   85.2  |
> | E-pointer             |  73.5  |   78.7  |
> | M-View                |  79.5  |   85.6  |
> | RE-Ext                |  78.6  |   85.4  |
> | Elastic               |  80.3  |   84.8  |
>
> + **Evaluating on complex structural equations**: We also analyzed the performance of our Expression-Tree decoding in comparison with Expression-Chain decoding (Seq2Exp baselines) in the context of complex equation scenarios. Table 5 showcases these results. In the first two simpler structures, Expression-Tree decoding did not lead to discernible differences. However, in the most complex third category of equations, our method (75.2\%) significantly outperformed M-View (65.7\%) and RE-Ext (64.7\%).
>
> + **Analyzing the impact of different query numbers**. We further analyze the impact of the number of queries on parallel decoding performance. The number of queries is set from 1 to 30. The results show as the number of queries increases, the performance initially increases notably and then decreases. Specifically, when there is only one query per layer (#query = 1), the parallel decoding strategy is removed. Conversely, when we adopt too many queries (#query >= 15), the performance of parallel decoding drops rapidly. We speculate that this might be because most of the queries are redundant and are matched to the "None" label under this case. Too many queries may lead to instability in training. Apart from these extreme cases (either too many or too few queries), the performance gains attributed to parallel decoding remain both stable and pronounced. For instance, as the number of queries fluctuates between 4 and 8, the improvement consistently remains within the range of +1.2\% to +1.5\%.
> |   # query         | 1   |   2   | 4   | 5   | 6   | 7   | 8   | 10  | 15  | 20  | 30  |
> |:----------------:|:---:|:---:|:---:|:---:|:---:|:---:|:---:|:---:|:---:|:---:|:---:|
> | Acc.(MathQA) %    | 79.9| 80.6| 81.4| 81.4| **81.5**| 81.3| 81.1| 80.8| 80.2| 79.8| 79.1|
> | Comp to #1       | 0   | +0.7| +1.5| +1.5| **+1.6**| +1.4| +1.2| +0.9| +0.3| -0.1| -0.8|
> ||||||||||||
> | Acc.( Math23K)   | 85.2| 85.8| **86.2**| 86.0| **86.2**| 85.9| 85.6| 85.2| 84.8| 83.5| 83.3|
> | Comp to #1       |  0  | +0.6| **+1**  | +0.8| **+1**| +0.7| +0.4|  0  | -0.4| -1.7| -1.9|
>
> These results demonstrate that our parallel decoding can exploit the structure of equations to construct expression trees, proving to be more suitable for equation generation than Expression-Chain.
>
>
> > ***Question B: What is the learnable query? How do it learn? How to predict expression?***
>
> Sorry for the confusion. We will make a clearer clarification regarding your concerns：
>
> + **What is the learnable query:** Learnable queries are essentially embedding vectors. Instead of denoting tokens or words with specific meanings, these queries serve as trainable feature detectors and learn to detect mathematical patterns or relations by attention mechanism in the transformer. After training, these feature detectors can extract specific mathematical relations from the question text. For instance, we observe that the #6 query typically matches division relations (Figure A4).
> + **How to predict:** The process of predicting the Left number, Right number, and Operator is carried out by feeding the learnable queries into three distinct classification heads. Each classification head is essentially a combination of an MLP (Multi-layer Perceptron) layer followed by a Softmax function. These heads output probabilities for each candidate value (l, op, r), and the one with the highest probability is chosen as the prediction (Equation 2-4).
> + **How do it learn:**  Analogous to how word embeddings in a vocabulary are trained, our learnable queries learn to detect and symbolize mathematical patterns or relations. To give an analogy, consider these queries as " band-pass filters", which are gradually specialized into "narrow-band filters" to detect specific mathematical patterns as the training progresses. The whole training process can also be seen as a denoising and restoration process. We start with **a noise token** (the randomly initialized query) and through training, optimize it to **a mathematical token** that represents specific mathematical expressions or relations.
>
>
> > ***Question C: How to handle those complex samples only with 8 transformer layers***
>
> + **Our parallel decoding mechanism can cover the vast majority of samples**. As each decoding layer can simultaneously output K expressions, our 8-layer model is capable of producing sufficiently complex expressions. According to our statistics, more than 94\% of the samples can be processed by our 8-layer model.
>
> + **To keep consistent with previous baselines**. Although increasing the decoding layers would allow our model to handle more samples, we also set the number of transformer layers to 8 to ensure a fair comparison with previous baselines (most Seq2Exp baselines set a maximum of 8-11 decoding steps).
>
> + **Our Variant model for equations of any length**. We acknowledge that certain extremely complex equations might exceed the processing capabilities of our standard model. To address this, we designed a variant model as shown in Figure A1 (Right). Here, a shared transformer block is used at each step. By repetitively feeding the queries and the problem representation into this block for K parallel expressions, our model has the potential to handle equations of any length. This layer-shared mechanism not only resolves the issue of equation length but has also achieved promising results in our experiments (Table 1, 2, 3).
>
>
> > ***Question D: Why do you need constants? How are they used? Will they affect performance?***
>
> Thank you for pointing out the use of numerical constants in our training and evaluation code. We'd like to clarify the rationale behind it and demonstrate its implications for assessment.
>
> + **Why do we need a constant vocabulary:** MWP is to translate human language into mathematical language (i.e. a finite set of symbols). Therefore, it is more efficient to design a specialized output vocabulary, rather than using the original vocabulary of a pre-trained model. Like most previous MWP approaches (GTS, G2T, RE-Ext, M-view, etc.), we set up the same constant vocabularies (Line 984-988). For instance, on Math23K, we defined two constant numbers: {1, $\pi$} as previous baselines. It is necessary to involve multiple mathematical operators and constant numbers in output vocabulary. The constant number is usually a common-sense number that doesn't appear in the question. For example, for a problem "What is the area of a circle with radius 3?". It cannot be solved without the constant " $\pi$" in the output vocabulary.
> + **How to use these constant numbers:** Each constant in the vocabulary is trained from scratch, i.e., First these constant embeddings are initialized randomly and are stored in the numerical vocabulary ($V_n$ in line 234), then are utilized for expression prediction (Equation 3) and subsequently updated during the training likes other parameters. Besides, we also experimented with initializing these constant vocabularies using the pre-trained embeddings from PLMs, but without any positive effect.
>
> + **Impact of these constants on evaluation:**  The use of constants enhances the model's capability to solve a wider array of problems. Without these constants, the model may struggle with specific problems, like those requiring the value of $\pi$. We remove the constant words and the corresponding samples and re-train the model, then test it on the entire test set. The accuracy of all methods dropped significantly:
>
> |               | Ours | Best Seq2tree (Ana-CL) | Best Seq2Exp (M-View) |
> |---------------|------|------------------------|-----------------------|
> | **w/ Constant**   | 86.2 | 85.6                   | 85.6                  |
> | **w/o Constant**  | 64.6 | 64.2                   | 61.9                  |
> | **Equation Case** |   (1/4 + 1) * 120 + 120   | Constant Number: 1  |  |
>
> Question Case: Given that a certain primary school has 120 male students and the number of female students is more than that of male students by (1/4), how many students are there in total in the school?
>
>
>
> > ***Question E: Compared with Self-Consistency in Table 3***
>
> Thank you for your thoughtful feedback. We will correct this mistake. Self-Consistency + GPT3.5 performs slightly better than our method on the MAWPS dataset (+0.2\%), but on the other two large datasets, Self-Consistency performs considerably worse than ours (-30.8\%, -20.1\%).
>
> > ***Improvements: some mistakes and the paper has more than 8 pages***
>
> Thank you for pointing out these grammatical mistakes. We will address these issues and add more diverse examples to demonstrate parallel decoding.
>
> Additionally, according to the official guidelines of EMNLP, the main manuscript is limited to 8 pages, followed by Limitations, References, and Appendix, which are of unlimited length. Our main text is less than 8 pages.
>
>
> --------------------------------------------------------------------------------------------------------------------------------------------------------------
> Thanks again for your valuable reviews above. If you have any further questions, please do not hesitate to let me know. I am more than happy to discuss with you (if the system allows).

---

### Meta-Review · Area_Chair_wWJv · 2023-09-19

**Recommendation:** 4

**Metareview:**

Summary: The paper focuses on generating the expression tree rather than the expression tokens sequentially for math equation generation. The proposed method combines the method from Seq2DAG and DETR, where Seq2DAG constructs the tree in a bottom-up layer-wise fashion and DETR uses a method to extract multiple objects in each layer in parallel. The system minimizes the number of steps involved in generating the expression structure by building the expression tree iteratively in bottom-up order. The system issues multiple queries to the problem representation at each step, generating valid expressions that are added to the problem embedding for the next step. The process stops when no valid expression is generated. The experimental results on the MWP task demonstrate the effectiveness of their approach, particularly for long equations and equations with complex tree structures.

Strengths: All the reviewers unanimously agree that this is a solid contribution with interesting implications. The proposed method adapts query-based detection combined with layer-wise decoding to math word problems seems novel and appropriate. The comprehensive experimentation conducted in this study greatly contributed to understanding the mechanism of the proposed method.

Weaknesses: I don't think there are any major weaknesses -- some of the weaknesses have been addressed during the discussion phase. The authors should take the comments during the rebuttal into account when preparing for the final version of the paper.

---

### Decision · Program_Chairs · 2023-10-07

**Decision:**

Accept-Main

**Comment:**

Summary: The paper focuses on generating the expression tree rather than the expression tokens sequentially for math equation generation. The proposed method combines the method from Seq2DAG and DETR, where Seq2DAG constructs the tree in a bottom-up layer-wise fashion and DETR uses a method to extract multiple objects in each layer in parallel. The system minimizes the number of steps involved in generating the expression structure by building the expression tree iteratively in bottom-up order. The system issues multiple queries to the problem representation at each step, generating valid expressions that are added to the problem embedding for the next step. The process stops when no valid expression is generated. The experimental results on the MWP task demonstrate the effectiveness of their approach, particularly for long equations and equations with complex tree structures.

Strengths: All the reviewers unanimously agree that this is a solid contribution with interesting implications. The proposed method adapts query-based detection combined with layer-wise decoding to math word problems seems novel and appropriate. The comprehensive experimentation conducted in this study greatly contributed to understanding the mechanism of the proposed method.

Weaknesses: I don't think there are any major weaknesses -- some of the weaknesses have been addressed during the discussion phase. The authors should take the comments during the rebuttal into account when preparing for the final version of the paper.